# Resilience, Positivity and Social Support as Perceived Stress Predictors among University Students

**DOI:** 10.3390/ijerph20196892

**Published:** 2023-10-07

**Authors:** Kamila Litwic-Kaminska, Aleksandra Błachnio, Izabela Kapsa, Łukasz Brzeziński, Jakub Kopowski, Milica Stojković, Darko Hinić, Ivana Krsmanović, Benedetta Ragni, Francesco Sulla, Pierpaolo Limone

**Affiliations:** 1Faculty of Psychology, Kazimierz Wielki University, 85-867 Bydgoszcz, Poland; alblach@ukw.edu.pl; 2Faculty of Political Science and Administration, Kazimierz Wielki University, 85-671 Bydgoszcz, Poland; izabela.kapsa@ukw.edu.pl; 3Faculty of Pedagogy, Kazimierz Wielki University, 85-064 Bydgoszcz, Poland; trener@ukw.edu.pl; 4Faculty of Computer Science, Kazimierz Wielki University, 85-064 Bydgoszcz, Poland; kopowski@ukw.edu.pl; 5Faculty of Technical Sciences, University of Kragujevac, 34000 Kragujevac, Serbia; milica.stojkovic@ftn.kg.ac.rs (M.S.); krsmanovici@gmail.com (I.K.); 6Department of Psychology, University of Kragujevac, 34000 Kragujevac, Serbia; dhinic@kg.ac.rs; 7Department of Humanistic Studies, Learning Science Hub, University of Foggia, 71122 Foggia, Italy; benedetta.ragni@unifg.it (B.R.); francesco.sulla@unifg.it (F.S.); pierpaolo.limone@unifg.it (P.L.)

**Keywords:** resilience, positivity, stress, university students, COVID-19 pandemic, mental health, academic concern

## Abstract

Since the COVID-19 pandemic, researchers have been trying to identify which personal resources can contribute to minimizing the mental health costs in students incurred due to the restrictions that disrupted safety and predictability in their academic lives. The aim of the study was to verify if and how individual factors (resilience and positivity) and socio-environmental factors (social support and nationality) allow prediction of the level of perceived stress. University students (*n* = 559) from Poland, Serbia, and Italy were surveyed using the Perceived Stress Scale (PSS-10), the Brief Resilience Scale (BRS), the Positivity Scale (PS), and the Interpersonal Support Evaluation List (ISEL-12). Personal resources—positivity, resilience, and support—were found to be positively interrelated and significantly associated with stress levels. Additionally, gender and nationality differentiated stress levels. A general linear model (GLM) showed that levels of perceived stress are best explained by resilience, positivity, tangible support, and gender. The results obtained can strengthen students’ awareness of personal resources and their protective role in maintaining mental health, as well as contribute to the creation of prevention-oriented educational activities. Nationality was not a significant predictor of the level of perceived stress, which highlights the universality of examined predictors among university students from different countries and suggests that interventions aimed at enhancing these resources could benefit students across different cultural contexts.

## 1. Introduction

COVID-19 must be considered a critical event in the educational biographies of the nearly 1.6 billion students who experienced the long-term closure of educational facilities [1]. For institutions, it has led to a hastily introduced technological revolution [2,3,4], shifting from the tradition of ‘face-to-face’ teaching to new ‘e-learning experiences’ hosted on digital platforms. These digital platforms, in turn, were perpetuated not by results, but by the extended duration of their use. Meanwhile, the students, as direct beneficiaries, have completely transitioned to distance learning in order to maintain the continuity of their studies. They had to cope with the acquisition of hardware and software to be able to participate in IT-based activities. For many, the COVID-19 lockdown resulted in a significant reduction or loss of their previous livelihoods. The hours spent in front of the computer caused an overload on the body (for many, this was associated with headaches and increased fatigue). Moreover, they were losing a sense of control over their lives and their educational process. This new and challenging situation, coupled with the fear of COVID-19, intensified their experience of stress [5]. Much academic effort has been dedicated to research investigating the extent of the mental and social health burden across various contexts. There is a current focus on finding resources that can reduce the costs associated with student ill-being (depression, anxiety, stress [5,6,7], and subsequent academic failure).

### 1.1. Defining and Measuring Resilience and Its Relations to the Perceived Stress

The COVID-19 pandemic has led to a situation that allows psychologists to observe competences and means that people employ in coping with adverse and even challenging conditions. The concept of resilience, introduced to psychology by Jack and Jeanne Block [8], refers to these competences.

Initially, resilience, seen as a set of fixed individual assets, was investigated through the use of psychometric questionnaires. One of these, the Dispositional Resilience Scale [9], was based on a theory of resilience-hardness encompassing three traits: commitment, control, and challenge [10]. Over time, both the definition of the concept and the measurement tools have evolved. The more recent and popular Resilience Scale [11] acknowledges resilience as an enduring characteristic of a person that manifests itself in two dimensions: personal competence and acceptance of self and life.

Then, a new perspective on resilience was introduced in The Connor–Davidson Resilience Scale (CD-RISC), which recognizes that the level of resilience is not constant, and may change depending on different experiences, states of health, and general functioning [12]. This questionnaire contains five factors: personal competence, spiritual influences, sense of internal control, affective tolerance, and acceptance of change, all of which are related to resilience.

Nowadays, the term resilience is no longer understood only as a feature of an individual (when it is used in this sense, the proper term is “resiliency”) [8,13,14,15] but as a process (in this case, the term “resilience” is used) [16,17,18,19]. Perceived as a process, resilience refers to efficiently overcoming difficult phenomena and events that affect a person throughout their life. Some researchers focus on critical phenomena, pointing out that resilience refers to a dynamic process of positive adaptation in the face of emerging adversities [20]. Others view resilience as a set of traits, an outcome, or a dynamic process that involves exposure to stress or adversity, followed by successful adaptation [21]. Despite the ambiguity of the terms, the literature commonly employs a broad definition of the term resilience [19,22,23].

In this study, we consider resilience as a process of positive adaptation to unfavorable conditions that are a source of stress [19,21] or as the “ability to cope with difficulties and recover from stress” [24]. The intertwined relationship between stress and resilience was more closely observed in the conditions of the pandemic and is well described in the quote: “Resilience is nurtured, developed, and mobilized in times of stress” [25]. In this sense, a resilient person is ready for positive adaptation and functioning despite prolonged exposure to stressors and disadvantages [26], such as the circumstances in the context of COVID-19.

### 1.2. The Concept of Positivity

Another concept that is being investigated in relation to stress is positivity [27,28,29]. In most research studies, the construct of positivity means possessing a positive outlook and openness to new and unfamiliar experiences [30]; it involves being stable in happiness despite environmental change [31] or adopting a favorable, hopeful, and confident perspective of the future, while considering adverse events as temporary [32]. It is also related to having optimistic, positive expectations about the future and holding positive views of self, others, the future, and life [33,34]. Some researchers identify positivity with positive thinking, indicating that it is at the core of individuals’ confidence in their future [35], while others describe it as a tendency to evaluate aspects of life in general as good [36].

Interest in positivity has been steadily growing over the past few decades and many studies have proven the relationship between positivity and such outcomes as mental health [37,38,39], physical health and hedonic balance [40,41], quality of relationship styles [42], positive affect [43], prosociality [44], and resilience [45]. In the context of COVID-19, some researchers have explored the correlation between positivity and fear or presented attitudes towards the circumstances and consequences of the pandemic [46,47,48], including among university students [49,50,51,52].

### 1.3. Definition and Measure of Social Support in the Relations to the Perceived Stress

Social support is an important health-related and communicative process, encompassing verbal and nonverbal behaviors aimed at assisting another person in need [53]. It is also categorized as perceived social support, meaning perceived availability and adequacy of supportive ties [54] or the resources that are provided by others [55]. Social support can relate to stress and stress outcomes in different ways: by directly affecting the occurrence of stress events, perceived stress, or stress outcomes [54,56,57].

The protective role of social support, recognized as a key resource in coping with stress, has attracted the interest of researchers, who have developed various scales to measure it. For example, the scale developed by Cohen and Hoberman [58] measures an individual’s perceived availability of social support and consists of four distinct subscales: tangible support, sense of belonging, self-esteem, and appraisal. This scale has been used for testing various groups of respondents, including students [59,60], and most recently during the COVID-19 pandemic [61].

### 1.4. In the Search for Cultural Differences in Students’ Experience of Stress

In emerging adulthood, it is crucial to connect with colleges and communities that contribute to well-being by supporting contexts and providing opportunities for social-emotional development [62]. However, the COVID-19 pandemic and the series of restrictions that accompanied it made this impossible [63]. Students, being a vulnerable population [64], were forced into long-term isolation. They feared for their physical health and expressed concerns for the continuity of their studies [7]. As a result, rates of depression, anxiety, stress, sleep disorders, and even suicide attempts have dramatically increased in the student population [1,65]. Ongoing research highlights the significant costs incurred by a large number of students as a result of the pandemic restrictions, which have disrupted safety and predictability in their academic lives. At the same time, researchers are trying to identify which personal resources can contribute to minimizing the costs of mental health and recovery [7].

It is worth examining in more detail to what extent personal resources facilitate students’ emotional balance among students of different cultures. Existing reports indicate cultural diversity in emotional functioning, personal characteristics and resources, and the way social support is given and perceived, along with other factors that influence the experience of stressful situations [66,67,68,69]. Although previous studies have analyzed stress levels among university students in different countries during the pandemic, revealing high levels of stress, anxiety, and depression, they were either conducted as a qualitative study aimed at understanding how pandemic experiences have affected student well-being [7] or the analyses were limited to one country with narrow study profiles [5,70]. In this study, we adopted a systemic approach, considering two perspectives that encompass individual factors (such as gender and personal resources including resilience and positivity) as well as socio-environmental factors (including social support and nationality). By conducting a cross-country study encompassing Poland, Serbia, and Italy, we aim to explore potential variations in the relationships between resilience, positivity, social support, and perceived stress across different cultural settings. This cross-cultural approach allows us to examine whether these relationships hold true universally or are influenced by cultural factors, thereby contributing to a more comprehensive understanding of the phenomenon. Consequently, by taking both perspectives into consideration (personal and socio-environmental), our goal was to examine the extent to which internal resources and external support influence the level of stress during periods of isolation and social distancing amid the COVID-19 pandemic across different cultures. Thus, the study aimed to verify if and how resilience, positivity, and social support enable us to predict the level of perceived stress in Polish, Serbian, and Italian students.

## 2. Materials and Methods

### 2.1. Participants and Procedure

The survey was conducted in June 2022 among Polish, Serbian, and Italian students studying at three universities: Kazimierz Wielki University (Bydgoszcz, Poland), University of Kragujevac (Serbia), and the University of Foggia (Italy). Universities were purposively selected for their size and their region-specific location (average multi-profile universities in medium-sized cities). The research was conducted fully anonymously, on a voluntary basis, and entirely online. Data were collected by three independent web applications (one for each country) to which individual access links were assigned. Inclusion criteria were as follows: active full-time and part-time student, at all three levels of education (bachelor’s, master’s, or PhD), under 40 years of age, studying at one of the designated universities. A balanced distribution of students from a range of technical and non-technical (humanities/social sciences, medical) subjects was also taken into account. All the respondents provided their informed consent to participate in the study. The study was conducted in accordance with the Declaration of Helsinki.

The sample consisted of 559 students in total.

The Polish sample was comprised of 306 students (189 females, 62%) pursuing bachelor’s degrees (78%) and master’s studies (22%), aged 18–39 (M = 22.56, SD = 3.13). Most of the participants were full-time students (87%) and studied outside their hometown (69%). Regarding the field of study, the sample presents the following distribution: social sciences 67%, exact and natural sciences 22%, health sciences 6%, and humanities 5%.

The Serbian sample consisted of 175 students (130 females, 74%), pursuing bachelor’s (85%), master’s (10%), and PhD degrees (5%), aged 18–38 (M = 22.21, SD = 3.03). Most of the participants were full-time students (86%) and studied outside their hometowns (62%). The majority of the participants were studying in the field of social sciences and humanities (38%), followed by medical disciplines (26%), IT (22%), and engineering and natural sciences and technology (14%).

The Italian sample comprised 78 students (44 females, 56%) pursuing bachelor’s (50%), master’s (27%), and PhD degrees (23%), aged 19–39 (M = 27.76, SD = 5.35). Most of the participants were full-time students (86%) and studied in their hometown (68%). The majority of the participants were studying in the field of natural and applied sciences (47%), followed by social sciences (36%), humanities (11%), and business (6%). The fields of study were determined on the basis of the regulations applied in each country.

### 2.2. Measures

The Perceived Stress Scale (PSS-10) [71] was used to evaluate the intensity of perceived stress experienced by the participants during the previous month. The responses for the 10 listed items are given on a five-point scale, ranging from “never” to “always”. The general result is calculated after reversing positive items’ scores and then summing up all scores. The total sum ranges from 0 (no stress) to 40 points (extreme stress). The reliability of the measure in all language versions was, respectively: Polish α = 0.85, Italian α = 0.89, Serbian α = 0.72.

The Brief Resilience Scale (BRS) [24] was implemented to measure resilience (i.e., the ability to overcome and/or recover from stress). It consists of six items rated on a five-point Likert-type scale (from 1 = strongly disagree, to 5 = strongly agree). The total score is calculated as a mean of the six items (after reversing three negative items). In the original study, the BRS demonstrated good internal consistency (with Cronbach’s alpha ranging from 0.80 to 0.91). In our study, it was, respectively: Polish α = 0.82, Italian α = 0.79, Serbian α = 0.80.

The Positivity Scale [34] was used to measure individuals’ tendency to see their lives and experiences with a positive outlook. It consists of eight items rated on a five-point Likert-type scale (from 1 = strongly disagree, to 5 = strongly agree). Item 4 (“At times, the future seems unclear to me”) was reverse-coded before running the statistical analyses. The score is the total score; the higher it is, the higher the level of positive orientation, and the range of raw scores is from 8 to 40. In the original study, the internal consistency was acceptable (Cronbach’s alpha ranging from 0.77 to 0.78). In our study, it was, respectively: Polish α = 0.91, Italian α = 0.88, Serbian α = 0.87.

Finally, the short form of the Interpersonal Support Evaluation List (ISEL-12) [72] was used as a measure of social support. The responses to the 12 items on a four-point scale ranging from “definitely false” to “definitely true” allow us to calculate a total score (range 0–36) that describes the overall perceived social support. The three subscales (scores range 0–12) indicate the perceived accessibility of appraisal (advice or guidance), belonging (empathy, acceptance, concern), and tangible social support (help or assistance, such as material or financial aid) [72]. The reliability of the measure in all samples was respectively: Polish α = 0.87, Italian α = 0.91, Serbian α = 0.87.

### 2.3. Data Analysis

The analyses were performed using Statistica v. 13 (StatSoft, Tulsa, OK, USA). The normal distribution of dependent variables was verified with Shapiro–Wilk’s test, as well as skewness and kurtosis values. National and gender differences were evaluated using ANOVA. Spearman R correlation was applied to determine the relationship between all continuous variables. A general linear model (GLM) was conducted to verify the relationship between gender, positivity, resilience, types of support as independent variables, nationality as a control variable, and the level of perceived stress as the dependent variable. GLM results were presented as standardized β coefficients (Beta) with 95% CI to determine which parameters were the strongest predictors of the dependent variable. The internal consistency of measures was estimated with Cronbach’s alpha coefficient. Statistical significance was defined for *p*-value lower than 0.05.

## 3. Results

### 3.1. Partial Correlations

The level of perceived stress was negatively correlated with resilience, positivity, and all types of support. All personal resources, positivity, resilience, and support, were positively interrelated (Table 1).

### 3.2. Descriptive Statistics

Descriptive statistics for the personal resources and the stress level in the groups of students from the three countries are presented in Table 2. There were significant national and gender differences in most of the personal resources and in the stress level. The level of stress and resilience varied across nationalities (*p* < 0.01) and genders (*p* < 0.001). In terms of stress level, Polish students scored significantly higher than Serbian students (*p* = 0.017), and women scored higher than men (*p* < 0.001). Polish students exhibited lower resilience compared to Italian students (*p* = 0.019), while women showed lower resilience than men (*p* < 0.001). Polish students also reported lower positivity compared to Serbian and Italian students (*p* < 0.001). The analysis revealed significant differences between males and females in all types of support, with women indicating a higher level of all kinds of support then men (*p* < 0.05).

Moreover, when comparing the averages of all groups with the norms [71], it was observed that students indicated rather high levels of stress. Additionally, our students reported lower resilience (*p* < 0.001) compared to the results obtained during the validation of the BRS by Smith et al. [24].

### 3.3. Perceived Stress Predictors

The GLM analysis revealed a significant model for the perceived stress: R2 = 0.48; F_(8, 546)_ = 62.22, *p* < 0.001. Resilience (ß = −0.39, *p* < 0.001) and positivity (ß = −0.38, *p* < 0.001) made the greatest contribution to the model. Nationality proved to be a non-significant factor (*p* > 0.05) (Table 3).

## 4. Discussion

Prior research has shown that students who are resilient successfully cope with the new and stressful situations of studying abroad, such as adjusting to a different university and in a different language [73]. Drawing from three different cultures—Serbian, Polish, and Italian—we sought to broaden the analysis of how resilience, positivity, and social support predict the level of stress experienced by students. Our results confirm that positivity, resilience, and support were positively interrelated and that they are significantly associated with stress levels. In our study, levels of perceived stress are best explained by resilience, positivity, tangible support, and gender. These findings are consistent with the results of previous studies that have emphasized the protective role of positivity against stress [27,37,43]. Some of these studies have proved that positivity plays an essential role in directly predicting the behaviors related to studying, learning, and performing under stress. Positivity is the dimension that enables maintenance of these competences despite the stress related to studying [50]. In another study, positivity was recognized as a significant factor supporting the use of self-regulated learning and COVID-19 psychosocial experience to cope with stress during the pandemic [51]. Additionally, other studies indicate that positivity is the most important predictor of emotional control (in terms of anger, depression, anxiety), allowing one to deal with these emotions [49], and resistance, hope and optimism are protective factors against depression [74], as well as a significant psychological source of strength, protecting against COVID-19-related mental issues [75].

International studies conducted among nursing students indicate that there are significant differences between nations in terms of depression and anxiety [76], mental health [77], as well as the level of anxiety suppression [49]. Similarly, in our study, we observed significant differences between students from different countries in terms of the level of stress and resilience. Notably, despite these differences, nationality was not a significant predictor of perceived stress. The resources such as resilience, positivity, and social support have a similar effect on stress in students regardless of nationality. While cultures can modify the experience of stress, their impact in the academic context may be tempered by the universalization of educational paths. In this context, the conducted research did not reveal critical patterns of different experiences of stress among young Serbs, Italians, or Poles. Future research should also explore the moderating and mediating role of other resource variables that reduce the direct and indirect negative impact of stress among students [78]. This underscores the universality of these factors in promoting resilience among university students and suggests that interventions aimed at enhancing these resources could benefit students across different cultural contexts. Similarly, our results are in line with those reported in the literature highlighting the significant role of resilience for student functioning [79,80,81]. We also further validated that resilience should be seen as a protective factor that reduces the level of perceived stress among students in all three educational contexts of Italy, Serbia, and Poland. These results are consistent with studies conducted on diverse student populations during the COVID-19 pandemic, which also demonstrated similar associations between resilience and stress levels [6,82,83,84,85]. Although, to our knowledge, there are no studies questioning the impact of resilience on coping with stress, there are researchers who propose a broader framework for understanding it. Padmanabhanunni et al. [86] perceive resilience as a protective factor that is beneficial for psychological health even if there is no stress [70].

The relationship between stress and other factors among students was also studied in relation to social support [82,87]. The results of this research indicate that social support is an important factor in coping with a stressful environment related to studying. However, the scope and forms of providing it at universities in different countries differ, as does the participation of students themselves in providing social support. Universities lack permanent strategies to legitimize and formalize student initiatives in this area, and the flow of information and popularization of good practices is limited.

Consistent with studies conducted in other countries during the pandemic [6,52,83,88,89], the respondents in our study reported high levels of stress and relatively low levels of resilience. Similarly to prior studies, higher levels of stress have also been identified among women [6,85,88], including the pre-pandemic period [90]. This highlights the importance of gender differences in understanding the stress levels among the student population.

Researchers have paid special attention to medical students as one of the most vulnerable groups in the context of COVID-19. Numerous studies showed that medical students demonstrated lower psychological resilience and higher perceived stress during the COVID-19 pandemic process [91,92,93,94,95]. Similar findings from students across various faculties in our study allow us to generalize these conclusions beyond the specific population. This suggests that the impact of the pandemic on students’ psychological well-being and coping abilities is a widespread phenomenon that extends beyond specific disciplines. However, on the other hand, it is worth revising the research framework so that, in addition to confirming the significant impact of resilience and positivity on distress, differences can also be seen. These differences may be conditioned by different approaches to studying, various access to resources, and current or hypothetical threats. For this reason, research needs to be replicated in different cultural, institutional, and social contexts.

### Study Limitations and Directions for Future Research

Our study was conducted in three countries and involved students from three universities with a wide range of fields of study. We are aware that undertaking a survey at one university in each country limits the possibility of generalizing the results obtained only to samples with similar characteristics. It would be worthwhile to extend further measurements to different types of universities (e.g., state and private, traditional, metropolitan, specialized, and profiled universities). On the other hand, we took care to ensure a variety of fields of study and a comparable distribution of students from different technical and non-technical (humanities/social sciences, medical) professions, assuming that they may have responded differently to the remote learning environment. This provided us with valuable data from different fields of study, at different stages of education. This provides a basis for capturing the actual needs of students with a realistic assessment of the representativeness of their needs and their intensity. However, this diversity is also a limitation of the study, as it requires a larger sample size, potentially utilizing purposive sampling based on the study field. Another limitation is the fact that the data were collected in the period just before the exam session, which may have elevated stress levels due to the impending exams. Furthermore, in addition to the usual stress (experienced before an exam), there was one related to the pandemic context of evaluation. Further research conducted at different times of the academic year may help clarify these factors. These cross-sectional studies are subject to well-known limitations: no inference to causality, reliance on self-report measures, lack of heterogeneity or small size of the sample [96]. Nevertheless, our study was guided by the main advantages of cross-sectional studies—they are relatively quick and inexpensive to conduct and allow us to collect data from a larger sample of participants and compare differences between groups, providing guidance for subsequent measurements. Thus, the cross-cultural research on resilience, positivity, social support, and perceived stress should take into account more contextual variables that take into account the specificity of individual countries (political, demographic, and economic conditions). Another disadvantage of cross-sectional measurements is that all measurements for an individual are taken at one point in time, despite recruiting the entire sample over a longer period. In our study, the measurement period was narrowed down as much as possible. All students were surveyed in the month before the end of the academic year, so during the credit and examination session. In the following studies, longitudinal data should be collected not only to determine possible casual relationships but also to monitor the effects of implementing the results of cross-cultural research in student intervention programs.

## 5. Conclusions

The findings of this study are of value in discussing the role of resilience, positivity, and social support in reducing perceived stress levels among university students. These results contribute to strengthening students’ awareness of their personal resources and their protective role in maintaining mental health. These data became helpful for our team in designing peer support training programs for university help seekers from Poland, Serbia, and Italy and to strengthen university policies aimed at involving students themselves in self-help. We believe these findings may also be useful for other practitioners designing cross-cultural educational campaigns for mental health prevention.

## Figures and Tables

**Table 1 ijerph-20-06892-t001:** Pearson R correlation matrix for the continuous variables.

	Resilience	Positivity	Appraisal Support	Belonging Support	Tangible Support
Resilience	1.00	0.46	0.20	0.25	0.20
Positivity	0.46	1.00	0.41	0.46	0.31
Appraisal support	0.20	0.41	1.00	0.63	0.64
Belonging support	0.25	0.46	0.63	1.00	0.58
Tangible support	0.20	0.31	0.64	0.58	1.00
Perceived stress	−0.57	−0.53	−0.22	−0.24	−0.24

Note: All relationships statistically significant at the *p* < 0.001 level.

**Table 2 ijerph-20-06892-t002:** Descriptive statistics of personal resources and stress in Polish, Serbian, and Italian students.

Nationality	Variable	M	SD	Range	Skewness	Kurtosis
Polish (*n* = 306)	Resilience	2.87	0.86	1–5	−0.03	−0.18
	Positivity	3.24	0.93	1–5	−0.32	−0.49
	Appraisal support	3.13	0.77	1–4	−0.58	−0.54
	Belonging support	2.92	0.77	1–4	−0.56	−0.17
	Tangible support	3.25	0.68	1–4	−0.73	0.09
	Perceived stress	23.16	7.19	2–40	−0.17	−0.33
Serbian (*n* = 175)	Resilience	3.02	0.75	1–5	0.20	−0.26
	Positivity	3.73	0.79	1–5	−0.69	0.23
	Appraisal support	3.07	0.78	1–4	−0.70	−0.10
	Belonging support	3.06	0.79	1–4	−0.64	−0.42
	Tangible support	3.22	0.63	1–4	−0.68	0.42
	Perceived stress	21.20	6.04	4–36	−0.02	−0.17
Italian (*n* = 78)	Resilience	3.22	0.72	1–5	−0.25	0.52
	Positivity	3.62	0.72	2–5	−0.29	0.12
	Appraisal support	3.06	0.80	1–4	−0.66	−0.05
	Belonging support	2.95	0.79	1–4	−0.49	−0.15
	Tangible support	3.05	0.80	1–4	−0.62	−0.43
	Perceived stress	21.94	7.37	3–38	−0.43	0.13

**Table 3 ijerph-20-06892-t003:** Results of the GLM predicting the level of stress by resilience, positivity, support, gender, and nationality group.

Variable		Beta	−95%; +95% CI	*t*	*p*
Resilience		−0.39	−0.46; −0.32	−10.70	0.000
Positivity		−0.38	−0.46; −0.30	−9.42	0.000
Appraisal support		0.01	−0.08; 0.10	0.21	0.837
Belonging support		0.09	0; 0.18	2.04	0.042
Tangible support		−0.12	−0.21; −0.04	−2.95	0.003
Nationality	Italian	0.08	0; 0.17	1.90	0.058
	Serbian	−0.09	−0.17; 0	−1.92	0.056
Gender	1	−0.15	−0.21; −0.09	−4.61	0.000

Note: We used the following coding for groups: 1 = males, 2 = females; CI = confidence interval for coefficients.

## Data Availability

The data presented in this study are available on request from the corresponding author.

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
