# Peer review of "Resilience, Positivity and Social Support as Perceived Stress Predictors among University Students"

_ijerph, 2023, doi:10.3390/ijerph20196892_

Round 1
Reviewer 1 Report
I am afraid that my overall impression of the manuscript is unfavorable because of my bias against cross-sectional analyses in the stress-support field. In this field, all associations are likely to embed reciprocal causality, which may not be tested via cross-sectional designs. Accordingly, my overarching position is that the field would be better of without publications resting on cross-sectional designs, with very rare exceptions pertaining to novel theoretical postulates (and this is not the case).
Reviewer 2 Report
The manuscript addresses a important aspect concerning the wellbeing of students. It is well written (though I still see a need a need for minor corrections of the language to improve on clarity, line 135-137), and the methods are clearly selected to match the objectives.
However, I have the following concerns:
1. The fact that ethical review and approval was waived in accordance with Polish law especially given that the study was not limited to Poland.
2. The discussion focuses on how the results obtained was consistent to results from other studies but I am wondering of there are other studies that have shown results that are not similar.
These concerns should be addressed in the revised version.
Kind regards
The manuscript addresses a important aspect concerning the wellbeing of students. It is well written (though I still see a need a need for minor corrections of the language to improve on clarity, line 135-137), and the methods are clearly selected to match the objectives.
However, I have the following concerns:
1. The fact that ethical review and approval was waived in accordance with Polish law especially given that the study was not limited to Poland.
2. The discussion focuses on how the results obtained was consistent to results from other studies but I am wondering of there are other studies that have shown results that are not similar.
These concerns should be addressed in the revised version.
Kind regards
Reviewer 3 Report
Comments and suggestions for Authors
I read with interest the manuscript entitled ‘Resilience, Positivity and Social Support as Perceived Stress Predictors among University Students’. The topic is interesting and it highlights the effects of the e-learning system due to Covid-19 pandemic. The authors analyze the level of perceived stress in academic environment and psychosocial factors (resilience, positivity and social support). I appreciate the authors for carrying out this research in three different countries, showing that increasing student resources by improving the mentioned factors is beneficial across different cultural contexts.
The article is well organized and contains all the sections. However, some revisions are needed.
Introduction
The subtitles are not entirely consistent with the content of the text. In the section 1.1. (Definitions and Concepts of Resilience and Its Relations to the Perceived Stress), there are information about different resilience scales. In the section 1.2. (Defining and Measuring Positivity), the notion of positivity is described, but there are no mentions about measuring it. In the section 1.3. (Social Support and Its Relations to the Perceived Stress) there is a paragraph about a scale which measures social support.
Authors should improve the Introduction. Psychological instruments are usually described in the Material and methods section, but not in the Introduction.
Lines 127 and 133 – there is the author’s name, it will have to be replaced with the corresponding number from the references list.
Materials and Methods
Authors should provide more information regarding the recruitment of study participants (the method of sampling, non-response rate, inclusion/ exclusion criteria, etc.)
I appreciate that the authors mentioned the internal consistency for the scales in the languages of the countries in which the study was carried out.
Results
Line 220-221 – ‘Polish students exhibited higher resilience compared to Italian students’, but the results from table 2 seem to show the opposite. The authors should check this out.
Discussions
The authors should improve this section. There are few paragraphs or ideas which:
- are not clearly formulated (lines 249-254);
- seem not to be related to the manuscript's themes (lines 262-267, about loneliness);
- contain formulation that does not reflect the results or conclusions of the cited study (lines 277-279) and may be interpreted differently (‘poorer results in psychological resilience’ is a negative aspect, but ‘poorer results in perceived stress and anxiety’ it means low levels of them, and that is a positive aspect).
11.07.2023
Reviewer 4 Report
I find it an interesting article because it explains the influence of stress on resilience, positivity, tangible support and gender. It is an interesting topic for the scientific community.
The sample of investigation is not representative It does not fully represent the study population. However, it should be specified that the study is a simple investigation of the variables studied as a first approach to the long-term research objective.
For future research I recommend using the instrument: Spielberger, C. D., Gorsuch, R. L., Lushene, R. E., & Cubero, N. S. (1999). STAI: State-Trait Anxiety Questionnaire. Madrid: TEA editions.
Researchers should significantly expand: “4. Discussion” and point “5. conclusions”.
I recommend citing studies from the same line of research that can be compared with the results obtained. This will allow the article to have a more international vision. Specifically the article: Levels of Stress, Anxiety, and Depression in University Students from Spain and Costa Rica during Periods of Confinement and Virtual Learning. https://doi.org/10.3390/educsci12100660
If the researchers make the indicated modifications, the article could be published.
